# GUARDIAN: DECOUPLING EXPLORATION FROM SAFETY IN REINFORCEMENT LEARNING

## ABSTRACT

Hybrid offline–online reinforcement learning (O2O RL) promises both sample efficiency and robust exploration, but suffers from instability due to distribution shift between offline and online data. We introduce RLPD-GX, a framework that decouples policy optimization from safety enforcement: a reward-seeking learner explores freely, while a projection-based guardian guarantees rule-consistent execution and safe value backups. This design preserves the exploratory value of online interactions without collapsing to conservative policies. To further stabilize training, we propose dynamic curricula that gradually extend temporal horizons and anneal offline–online data mixing. We prove convergence via a contraction property of the guarded Bellman operator, and empirically show state-of-the-art performance on Atari-100k, achieving a normalized mean score of 3.02 (+45% over prior hybrid methods) with stronger safety and stability. Beyond Atari, ablations demonstrate consistent gains across safety-critical and long-horizon tasks, underscoring the generality of our design. Extensive and comprehensive results highlight decoupled safety enforcement as a simple yet principled route to robust O2O RL, suggesting a broader paradigm for reconciling exploration and safety in reinforcement learning.

## 1 INTRODUCTION

Deep reinforcement learning (DRL) has demonstrated remarkable performance in complex decision-making tasks such as strategy games and robotic control (Mnih et al., 2015; Arulkumaran et al., 2017; Li, 2018; Dulac-Arnold et al., 2021). However, its mainstream paradigms, i.e., purely online learning and purely offline learning, are constrained by sample inefficiency and out-of-distribution (OOD) generalization challenges, respectively (Fujimoto et al., 2019; Kumar et al., 2020; Gu et al., 2024b). To address these issues, Offline-to-Online (O2O) reinforcement learning introduces a two-stage paradigm, (Gulcehre et al., 2021; Sönmez et al., 2024; Figueiredo Prudencio et al., 2024) where the agent is first pretrained on offline data and then fine-tuned online, thereby alleviating the weaknesses of both approaches. While effective in principle, this rigid two-stage design often exacerbates distributional shifts, induces compounded Bellman errors, and causes performance regressions (Figueiredo Prudencio et al., 2024; Shakya et al., 2023). Recent research has thus moved toward integrated training loops, where offline data serve as a regularizing prior to guide safe exploration and suppress overgeneralization, while online samples immediately correct value overestimation caused by incomplete offline coverage (Shin et al., 2025; Niu et al., 2023). This synergy ensures a smooth transition between exploration and exploitation, yielding more stable and efficient performance improvements.

Nevertheless, hybrid offline-online reinforcement learning in practice often struggles to reconcile the mismatch between the behavior policy underlying offline trajectories and the evolving target policy of the agent (Wen et al., 2024; Sönmez et al., 2024). This distribution gap leads to overly conservative behaviors, where the model performs well near the offline distribution but fails to explore new actions; to over-

optimism, where values are overestimated in out-of-distribution regions; and to training oscillations, where conflicting learning signals undermine convergence (Figueiredo Prudencio et al., 2024; Chen et al., 2023). Although prior approaches, such as RLPD and Hy-Q attempt to mitigate this issue, they remain limited. Specifically, both rely on injecting strong conservative biases: RLPD (Ball et al., 2023b) constrains exploration strictly within the offline distribution at the policy level, while Hy-Q (Song et al., 2023)systematically underestimates values of unknown actions at the critic level. Despite their different mechanisms, both approaches converge to the same dilemma: in order to stabilize the transition phase, they suppress the exploratory value of online data, leading to suboptimal policies that remain tethered to the offline distribution and fail to fully exploit the potential of online interaction.

To overcome this limitation, we propose the RLPD-GX framework, whose central contribution lies in decoupling policy learning from safety enforcement: a constraint-free *Learner* is responsible for exploration and reward maximization, while a *Guarded Bellman Operator* projects online actions onto a predefined safe subspace to guarantee verifiable execution. This design preserves the intrinsic exploratory value of online interactions, while effectively filtering out spurious signals arising from random exploration that could misguide policy updates. To ensure a smooth transition from offline pretraining to online fine-tuning, our RLPD-GX further introduces dynamic curriculum sampling: (i) **Dynamic Temporal Sampling (DTS)** establishes a temporal curriculum that transitions from dense (Narvekar et al., 2020; Portelas et al., 2020), short-horizon sampling to sparse, long-horizon sampling, thereby balancing local rule learning with long-term planning; and (ii) **Dynamic Symmetric Sampling (DSS)** smoothly adjusts the mixing ratio between offline and online data, starting with an offline-biased phase to distill prior knowledge and converging to a balanced 1:1 mixture, thereby avoiding conflicts and instabilities. This framework fundamentally reshapes the relationship between safety and optimality by decoupling the two, turning the pursuit of optimal policies under safety constraints from a zero-sum trade-off into a feasible, synergistic goal.

Extensive experiments are conducted to validate these claims. First, on the challenging Atari 100k benchmark (Ye et al., 2021), we demonstrate that RLPD-GX achieves superior performance and sample efficiency compared to state-of-the-art online, offline, and hybrid baselines. Second, we perform a targeted analysis showing that our decoupled Guardian mechanism provides stronger safety guarantees and higher task returns than representative safe RL algorithms. Finally, a series of ablation studies confirms that the proposed dynamic sampling mechanisms are critical for achieving faster and more stable convergence. These results set a new benchmark, showing our design breaks the safety–performance trade-off.

## 2 PROBLEM FORMULATION

We formalize the problem of safe reinforcement learning within a hybrid offline-online data regime. Our formulation is grounded in the established framework of Markov Decision Processes (MDPs) (Puterman, 1994; Gu et al., 2024b; White, 1993), extended to accommodate externally specified safety constraints and a composite data stream.

### 2.1 MDPs IN A HYBRID DATA REGIME

We model the environment as a **Markov Decision Process (MDP)**, defined by the tuple $(\mathcal{S}, \mathcal{A}, P, R, \gamma)$, representing the state space, action space, transition probability function $P : \mathcal{S} \times \mathcal{A} \times \mathcal{S} \rightarrow [0, 1]$, a bounded reward function $R : \mathcal{S} \times \mathcal{A} \rightarrow \mathbb{R}$, and a discount factor $\gamma \in [0, 1)$. The agent's learning process is fueled by a **hybrid data stream**. Let $d_{\text{off}}(s, a)$ be the state-action marginal distribution of the static **offline dataset** $\mathcal{D}_{\text{off}}$, and let $d_{\text{on}}(s, a)$ be the corresponding distribution for the dynamically populated **online replay buffer** $\mathcal{B}_{\text{on}}$. The composite training distribution $d_{\text{train}}$ from which data is sampled is a time-varying convex combination:

$$d_{\text{train}}(s, a; t) = \lambda(t)d_{\text{on}}(s, a) + (1 - \lambda(t))d_{\text{off}}(s, a) \tag{1}$$

where $\lambda(t) \in [0, 1]$ is a mixing coefficient at training step $t$, for which we propose a dynamic annealing schedule (detailed in Section 3.2). The primary theoretical challenge arises from the distributional shift between $d_{\text{off}}$ and the distribution induced by the evolving online policy $d^{\pi_\phi}$. This shift can lead to severe extrapolation errors and value overestimation for out-of-distribution (OOD) actions.

## 2.2 SAFETY AS A STATE-ACTION CONSTRAINT

We depart from safe RL formulations that integrate safety as a penalty or cost. Instead, we define safety as a **hard constraint** on the policy's support. This is enforced by an external, deterministic predicate $g : \mathcal{S} \times \mathcal{A} \rightarrow \{0, 1\}$, where $g(s, a) = 1$ signifies that action $a$ is permissible in state $s$.

This predicate defines a state-dependent safe action set:

$$\mathcal{A}_{\text{safe}}(s) \triangleq \{a \in \mathcal{A} \mid g(s, a) = 1\} \tag{2}$$

Consequently, the space of valid policies is constrained to $\Pi_{\text{safe}}$, where

$$\Pi_{\text{safe}} = \{\pi \in \Pi \mid \text{supp}(\pi(\cdot|s)) \subseteq \mathcal{A}_{\text{safe}}(s), \forall s \in \mathcal{S}\} \tag{3}$$

where $\Pi$ is the set of all possible stochastic policies. This reframes the problem as a constrained optimization task rather than a multi-objective one.

## 2.3 THE MAXIMUM ENTROPY LEARNING OBJECTIVE

The agent's goal is to find a policy $\pi_\phi \in \Pi$ that maximizes the **maximum entropy objective**. This objective encourages exploration and improves robustness by seeking both high returns and high policy entropy:

$$J(\pi_\phi) = \mathbb{E}_{\substack{s_t \sim \rho_{\pi_\phi} \\ a_t \sim \pi_\phi(\cdot|s_t)}} \left[ \sum_{t=0}^{\infty} \gamma^t \left( R(s_t, a_t) + \alpha \mathcal{H}(\pi_\phi(\cdot \mid s_t)) \right) \right], \tag{4}$$

where $\rho_{\pi_\phi}$ is the state distribution induced by policy $\pi_\phi$, and $\mathcal{H}$ is the Shannon entropy. The corresponding soft Q-function, $Q^*(s, a)$, is the unique fixed point of the soft Bellman operator $\mathcal{T}^{\text{soft}}$:

$$(\mathcal{T}^{\text{soft}} Q)(s, a) = R(s, a) + \gamma \mathbb{E}_{s' \sim P(\cdot|s,a)} \left[ V_{\text{soft}}(s') \right], \tag{5}$$

where the soft value function is $V_{\text{soft}}(s') = \mathbb{E}_{a' \sim \pi_\phi(\cdot|s')}[Q(s', a') - \alpha \log \pi_\phi(a'|s')]$. Our methodology adapts this operator to respect the safety constraints defined in Eq. 2. This adaptation is realized through our **Guarded Backup** mechanism for practical value updates (Section 3.2.1) and formalized by the **Guarded Bellman Operator** for theoretical analysis (Section 3.3). We formally prove that this adapted operator maintains the crucial contraction property, guaranteeing convergence, in Appendix B.

## 3 METHODOLOGY: DECOUPLING LEARNING FROM SAFETY ENFORCEMENT

The cornerstone of RLPD-GX is the **decoupling of policy optimization from safety enforcement**. This principle avoids the complexities of multi-objective optimization, where conflicting gradients for reward maximization and constraint satisfaction can destabilize training. Instead, we structure the problem as a constrained optimization task solved via a projection-based method (Wachi & Sui, 2020; Chow et al., 2017). This consists of two orthogonal components: a reward-seeking **Learner** and a safety-enforcing **Guardian**.

## 3.1 SYSTEM ARCHITECTURE AND DATA FLOW

At each timestep $t$, the Learner's unconstrained policy $\pi_\phi$ proposes a raw action $a_t \sim \pi_\phi(\cdot|s_t)$. The Guardian module then projects this action onto the safe set defined in Eq. 2 to produce a certified action $a_t^{\text{exec}}$:

$$a_t^{\text{exec}} = \Pi_{\text{safe}}(s_t, a_t) := \arg \min_{a' \in \mathcal{A}_{\text{safe}}(s_t)} \|a' - a_t\|_2^2. \tag{6}$$

Figure 1: Architecture of **RLPD-GX**. A Learner explores freely, while a projection-based Guardian ensures safe execution and guarded value backups. Dynamic sampling (DTS/DSS) with OOD regularization stabilizes hybrid offline–online learning, enabling safe yet exploratory policy updates.

Only $a_t^{\text{exec}}$ is executed. This ensures the behavior policy, i.e., the policy generating the online data, is always within $\Pi_{\text{safe}}$, while the Learner's policy $\pi_\phi$ can maintain its full expressive capacity. The resulting transition $(s_t, a_t^{\text{exec}}, r_t, s_{t+1})$ is guaranteed to be safe and is stored in $\mathcal{B}_{\text{on}}$. The Learner thus optimizes on a sanitized data stream, eliminating any direct exposure to unsafe actions and their consequences. The complete training procedure is summarized in Algorithm 1.

### 3.2 THE LEARNER: PRINCIPLED OPTIMIZATION ON HETEROGENEOUS DATA

The Learner is designed for stable and efficient optimization, addressing the challenges of heterogeneous data through three key mechanisms.

**(a) Dynamic Temporal Sampling (DTS)**    To mitigate high variance in initial learning stages, DTS implements a curriculum over the temporal structure of sampled data. By initially prioritizing short, contiguous sequences, DTS provides low-variance gradient estimates for learning local dynamics. The sampling interval $\Delta(t)$ gradually expands:

$$\Delta(t) = \Delta_{\min} + (\Delta_{\max} - \Delta_{\min}) \cdot \left(\frac{t}{T}\right)^\beta. \tag{7}$$

This allows the agent to build a foundation of basic behaviors before tackling long-term credit assignment, promoting a more stable convergence trajectory.

**(b) Dynamic Symmetric Sampling (DSS)**    To manage the non-stationary nature of the training distribution, DSS provides a distributional annealing schedule. It smoothly varies the mixing parameter $\lambda(t)$ from the hybrid data distribution (Eq. 1) for online data:

$$\lambda(t) = \lambda_{\min} + (\lambda_{\max} - \lambda_{\min}) \cdot \sigma\left(k \cdot \left(t - \frac{T}{2}\right)\right). \tag{8}$$

This prevents abrupt shifts in the data landscape, allowing the function approximators to adapt gradually from offline knowledge distillation to online refinement.

#### 3.2.1 GUARDED BACKUPS FOR CONSISTENT VALUE LEARNING

For the value function to be consistent with the actual execution policy, Bellman backups must only consider safe actions. We first define a safe policy distribution, $\pi_\phi^{\text{safe}}$, by re-normalizing $\pi_\phi$ over the safe action set

$\mathcal{A}_{\text{safe}}(s')$ (defined in Eq. 2):

$$\pi_\phi^{\text{safe}}(a'|s') = \frac{\pi_\phi(a'|s') \cdot \mathbb{I}(a' \in \mathcal{A}_{\text{safe}}(s'))}{\sum_{a'' \in \mathcal{A}_{\text{safe}}(s')} \pi_\phi(a''|s')}. \tag{9}$$

where $\mathbb{I}(\cdot)$ is the indicator function. The target value $y$ for a transition $(s, a, r, s')$ is then constructed using this safe policy:

$$y = r + \gamma \left( \mathbb{E}_{a' \sim \pi_\phi^{\text{safe}}(\cdot|s')}[Q_{\min}(s', a')] - \alpha \mathcal{H}(\pi_\phi^{\text{safe}}(\cdot \mid s')) \right) \tag{10}$$

where $Q_{\min}$ is the pessimistic estimate from a conservative Q-ensemble. This guarded target ensures that the Learner's value estimates align with the outcomes of the Guardian's safety enforcement.

## 3.3 THEORETICAL FOUNDATION: CONVERGENCE OF GUARDED VALUE ITERATION

A critical theoretical question is whether the introduction of the Guardian's projection preserves the convergence properties of value-based reinforcement learning. We demonstrate that it does by defining a Guarded Bellman Operator and proving it is a contraction mapping.

**Definition 1** (Guarded Bellman Operator). *For any Q-function $Q : \mathcal{S} \times \mathcal{A} \to \mathbb{R}$, the Guarded Bellman Operator $\mathcal{T}_\Pi$ is a mapping from $Q$ to $\mathcal{T}_\Pi Q$ such that for any state-action pair $(s, a)$, the maximization is performed over the safe action set from Eq. 2:*

$$(\mathcal{T}_\Pi Q)(s, a) \triangleq R(s, a) + \gamma \mathbb{E}_{s' \sim P(\cdot|s,a)} \left[ \max_{a' \in \mathcal{A}_{safe}(s')} Q(s', a') \right] \tag{11}$$

**Theorem 1** (Contraction). *The operator $\mathcal{T}_\Pi$ is a $\gamma$-contraction in the max norm $\|\cdot\|_\infty$.*

*Proof Sketch.* Let $Q_1$ and $Q_2$ be two arbitrary Q-functions. We examine the max-norm distance between their mappings under $\mathcal{T}_\Pi$:

$$\|\mathcal{T}_\Pi Q_1 - \mathcal{T}_\Pi Q_2\|_\infty = \max_{s,a} |(\mathcal{T}_\Pi Q_1)(s, a) - (\mathcal{T}_\Pi Q_2)(s, a)|$$

$$= \max_{s,a} \left| \gamma \mathbb{E}_{s'} \left[ \max_{a' \in \mathcal{A}_{\text{safe}}(s')} Q_1(s', a') - \max_{a' \in \mathcal{A}_{\text{safe}}(s')} Q_2(s', a') \right] \right|$$

$$\leq \gamma \max_{s,a} \mathbb{E}_{s'} \left| \max_{a' \in \mathcal{A}_{\text{safe}}(s')} Q_1(s', a') - \max_{a' \in \mathcal{A}_{\text{safe}}(s')} Q_2(s', a') \right|$$

$$\leq \gamma \max_{s'} \max_{a'' \in \mathcal{A}_{\text{safe}}(s')} |Q_1(s', a'') - Q_2(s', a'')|$$

$$\leq \gamma \max_{s',a'} |Q_1(s', a') - Q_2(s', a')| = \gamma \|Q_1 - Q_2\|_\infty$$

The key step relies on the property that $|\max_{x \in X} f(x) - \max_{x \in X} g(x)| \leq \max_{x \in X} |f(x) - g(x)|$. Since $\gamma < 1$, the operator is a contraction. $\square$

**Implication** By the Banach fixed-point theorem, Theorem 1 ensures that repeated application of $\mathcal{T}_\Pi$ converges to a unique fixed point $Q_\Pi^*$, the optimal Q-function for the safety-constrained MDP. This result shows our decoupled framework optimizes toward a well-defined, provably safe value function with preserved convergence guarantees. The complete proof is provided in Appendix B.

---

**Algorithm 1** RLPD-GX: Decoupled Learning and Safety Enforcement

---

1: **Initialize:** Learner policy $\pi_\phi$, Q-function ensemble $\{Q_{\theta_i}\}_{i=1}^N$, target networks $\{Q_{\theta'_i}\}_{i=1}^N$.
2: **Initialize:** Offline dataset $\mathcal{D}_{\text{off}}$, empty online replay buffer $\mathcal{B}_{\text{on}}$.
3: **Initialize:** Safety predicate function $g(s,a)$ to define $\mathcal{A}_{\text{safe}}(s)$ from Eq. 2.
4: **for** training step $t = 1, \ldots, T$ **do**
5:                                 ▷ *— Online Interaction Phase (Guardian Enforces Safety) —*
6:     Observe current state $s_t$.
7:     Learner proposes a raw action: $a_t \sim \pi_\phi(\cdot|s_t)$.
8:     Guardian projects to a safe action: $a_t^{\text{exec}} \leftarrow \arg\min_{a' \in \mathcal{A}_{\text{safe}}(s_t)} \|a' - a_t\|_2^2$.
9:     Execute $a_t^{\text{exec}}$, observe reward $r_t$ and next state $s_{t+1}$.
10:    Store sanitized transition $(s_t, a_t^{\text{exec}}, r_t, s_{t+1})$ in online buffer $\mathcal{B}_{\text{on}}$.
11:
12:                             ▷ *— Learner Update Phase (Learner Optimizes Policy) —*
13:    Update DSS mixing parameter $\lambda(t)$ and DTS sampling interval $\Delta(t)$.
14:    Sample minibatch $\mathcal{B}_{\text{off}} \sim \mathcal{D}_{\text{off}}$ and $\mathcal{B}_{\text{on}} \sim \mathcal{B}_{\text{on}}$ according to $\lambda(t)$ and $\Delta(t)$.
15:    Form combined batch $\mathcal{B} \leftarrow \mathcal{B}_{\text{off}} \cup \mathcal{B}_{\text{on}}$.
16:                                  ▷ *Calculate Guarded Backup Target*
17:    For each $(s,a,r,s')$ in $\mathcal{B}$, define safe policy $\pi_\phi^{\text{safe}}(\cdot|s')$ by re-normalizing $\pi_\phi(\cdot|s')$ over $\mathcal{A}_{\text{safe}}(s')$.
18:    Compute target value $y$ using pessimistic target Q-ensemble $Q'_{\min} = \min_i Q_{\theta'_i}$:

$$y \leftarrow r + \gamma \left( \mathbb{E}_{a' \sim \pi_\phi^{\text{safe}}(\cdot|s')}[Q'_{\min}(s', a')] - \alpha \mathcal{H}(\pi_\phi^{\text{safe}}(\cdot \mid s')) \right) \qquad \triangleright \text{Eq. 10}$$

19:                                      ▷ *Update Critic (Q-functions)*
20:    Update each Q-function $Q_{\theta_i}$ by minimizing soft Bellman error: $\mathcal{L}_{Q_i} = \mathbb{E}_\mathcal{B}\left[(Q_{\theta_i}(s,a) - y)^2\right]$.
21:                                       ▷ *Update Actor (Policy)*
22:    Update policy $\pi_\phi$ via: $\mathcal{L}_\pi = \mathbb{E}_{s \sim \mathcal{B}, a \sim \pi_\phi}[\alpha \log(\pi_\phi(a|s)) - \min_i Q_{\theta_i}(s,a)]$.
23:                                       ▷ *Update Target Networks*
24:    Update target Q-networks softly: $\theta'_i \leftarrow \tau\theta_i + (1-\tau)\theta'_i$ for all $i$.

---

## 4 EXPERIMENTS

### 4.1 MAIN COMPARISONS AND ANALYSES

**Experiment Setup.** We evaluate our method RLPD-GX on the Atari 100k benchmark, the gold standard for assessing sample efficiency in reinforcement learning. It comprises 26 diverse games, challenging agents to learn effective policies within a strict budget of 100,000 environment steps (400k frames). The explicit rule-based constraints in Atari games, such as life penalties, make it an ideal platform for studying safety and constraint adherence. Furthermore, the availability of official offline datasets, such as RL Unplugged, facilitates research across offline, online, and offline-to-online (O2O) paradigms. We compare **RLPD-GX** against three categories of representative baselines: **Offline learning** (**JOWA** (Cheng et al., 2025) with adaptive replay; **EDT** (Wu et al., 2023), a model-based approach), **Online learning** (**STORM** (Zhang et al., 2023a), goal-oriented with a transitive model; **DreamerV3** (Hafner et al., 2024a), world-model based; **DramaXS** (Wang et al., 2025), exploration-focused; **BBF** (Schwarzer et al., 2023a), value-gradient based; **EZ-V2** (Wang et al., 2024a), simplified and efficient), and **Hybrid learning (O2O)** (**RLPD** (Ball et al., 2023a), distributed training **MuZero Unplugged** (Schrittwieser et al., 2021a), combining model-based and model-free).

**Main Comparisons.** On the **Atari 100k benchmark**, RLPD-GX achieves a normalized mean of **3.02**, clearly surpassing offline (**EDT 2.39, JOWA 2.35**), online (**DreamerV3 1.27, STORM 1.27**), and hybrid baselines (**MuZero Unplugged 1.97, RLPD 2.07**), demonstrating superior sample efficiency. The **primary**

| Game | Random | Human | RLPD | MuZero Unplugged | JOWA | EDT | STORM | DreamerV3 | DramaXS | BBF | EZ-V2 | RLPD-GX |
|---|---|---|---|---|---|---|---|---|---|---|---|---|
| Alien | 228 | 7128 | 1264 | 746 | 1726 | 1664 | 984 | 959 | 820 | 1173 | 1558 | **2365** |
| Amidar | 6 | 1720 | 162 | 76 | 215 | 102 | 205 | 139 | 131 | 245 | 185 | **286** |
| Assault | 222 | 742 | 1826 | 643 | **2302** | 1624 | 801 | 706 | 539 | 2091 | 1758 | 2136 |
| Asterix | 210 | 8503 | 1864 | 29062 | 9624 | 11765 | 1028 | 932 | 1632 | 3946 | **61810** | 23672 |
| Bank Heist | 14 | 753 | 543 | 593 | 32 | 14 | 641 | 649 | 137 | 733 | 1317 | **1582** |
| BattleZone | 2360 | 37188 | 16240 | 11286 | 18627 | 17540 | 13540 | 12250 | 10860 | 24460 | 14433 | 20326 |
| Boxing | 0 | 12 | 72 | 62 | 89 | 82 | 80 | 78 | 78 | 86 | 75 | **95** |
| Breakout | 2 | 30 | 426 | 390 | 376 | 235 | 16 | 31 | 7 | 371 | 400 | **562** |
| ChopperCommand | 811 | 7388 | 2346 | 1764 | 3813 | 3576 | 1888 | 420 | 1642 | **7549** | 1197 | 4624 |
| CrazyClimber | 10780 | 35829 | 87264 | 93268 | 97682 | 114253 | 66776 | 97190 | 83931 | 58432 | 112363 | **124632** |
| DemonAttack | 152 | 1971 | 7628 | 8496 | 3548 | 21752 | 165 | 303 | 201 | 13341 | 22774 | 12362 |
| Freeway | 0 | 30 | 21 | 23 | 18 | 25 | 34 | 0 | 15 | 26 | 0 | **36** |
| Frostbite | 65 | 4335 | 3726 | 4051 | 1824 | 2164 | 1316 | 909 | 785 | 2385 | 1136 | **4264** |
| Gopher | 258 | 2412 | 2342 | 2640 | **8460** | 7635 | 8240 | 3730 | 2757 | 1331 | 3869 | 4624 |
| Hero | 1027 | 30826 | 6372 | 4326 | 12476 | **17645** | 11044 | 11161 | 7946 | 7819 | 9705 | 7426 |
| Jamesbond | 29 | 303 | 756 | 602 | 864 | 642 | 509 | 445 | 372 | 1130 | 468 | **1276** |
| Kangaroo | 52 | 3035 | 6836 | 4326 | 7642 | **8970** | 4208 | 4098 | 1384 | 6615 | 1887 | 6824 |
| Krull | 1598 | 2666 | 8924 | 5673 | 9230 | 8624 | 8413 | 7782 | 9693 | 8223 | 9080 | **9762** |
| KungFuMaster | 258 | 22736 | 15264 | 20326 | 18624 | 16462 | 26183 | 21420 | 23920 | 18992 | **28883** | 24382 |
| MsPacman | 307 | 6952 | 3869 | 4539 | 1962 | 2370 | 2673 | 1327 | 2270 | 2008 | 2251 | **5024** |
| Pong | -21 | 15 | 14 | 18 | 19 | 14 | 11 | 18 | 15 | 17 | 21 | 21 |
| PrivateEye | 25 | 69571 | 4632 | 3726 | 302 | 162 | **7781** | 882 | 90 | 41 | 100 | 4236 |
| Qbert | 164 | 13455 | 10624 | 13121 | 13260 | 11735 | 4522 | 3405 | 796 | 4447 | 16058 | **17464** |
| RoadRunner | 12 | 7845 | 18262 | 32460 | **46240** | 36574 | 17564 | 15565 | 14020 | 33427 | 27517 | 38296 |
| Seaquest | 68 | 42055 | 672 | 6745 | 2725 | 3762 | 525 | 618 | 497 | 1233 | 1974 | **7245** |
| UpNDown | 533 | 11693 | 10264 | 6432 | **16270** | 13287 | 7985 | 7667 | 7387 | 12102 | 15224 | 10382 |
| Normalised Mean (%) | 0 | 1 | 2.07 | 1.97 | 2.35 | 2.39 | 1.27 | 1.12 | 1.05 | 2.26 | 2.69 | **3.02** |
| Normalised Median (%) | 0 | 1 | 0.82 | 0.91 | 1.05 | 0.92 | 0.58 | 0.49 | 0.27 | 0.92 | 1.23 | **1.25** |

Table 1: Atari 100k benchmark results. **RLPD-GX** consistently outperforms offline, online, and hybrid baselines across 26 games, achieving the best normalized mean (3.02) and median (1.25) scores.

**driver** of these gains is the rule-consistent **safety enforcement mechanism**, which projects actions into the safe subspace during execution and value backups, ensuring valid online data. This yields marked advantages in **safety-critical tasks** (e.g., **Seaquest 7245** vs. **EDT 3762**, **DreamerV3 525**; **PrivateEye 4236** vs. **MuZero 3726**, **EDT 162**) and in **complex environments** (e.g., **BattleZone 20326**, **Qbert 17464**), consistently outperforming baselines. **Dynamic sampling** further aids **long-horizon tasks**, such as **Frostbite (4264)** vs. **EDT (2164)**, **DreamerV3 (909)**, and **Krull (9762)** vs. **MuZero (5673)**, **DreamerV3 (7782)**. The normalized median of **1.25** confirms that **safety enforcement is the decisive factor**, with sampling providing stability in temporally extended settings.

### 4.2 EFFICACY ANALYSES OF THE SAFETY GUARD MECHANISM

To validate the effectiveness of our proposed **Guardian (G0)** framework in mitigating the distributional shift between offline priors and online interactive data, we conduct a comparative evaluation against four representative baselines in the RLPD SEAQUEST environment: (G1) *No Guard* (Stolz et al., 2024), (G2) *Execution Mask Only*, (G3) *CMDP-Lagrangian* (Wang et al., 2023), and (G4) *Classifier Shield* (Yang et al., 2023). During training, we track the **temporal-difference (TD) error** and **Q-ensemble variance** to assess the stability and epistemic uncertainty of the learning process as it integrates offline knowledge. After training, we further conduct a **critical state replay** evaluation to examine the robustness of safety policies generalized from offline data when faced with previously unseen critical boundaries. This evaluation comprises two controlled tests: (i) **margin scanning**, which probes decision accuracy under boundary perturbations, and (ii) **full-episode replay**, which assesses long-term safety durability by measuring the *Time-To-First Violation (TTFV)*. As shown in Figure 2, **Guardian (G0)** demonstrates comprehensive superiority over all baselines in both stabilizing offline-to-online learning and ensuring safety. During training, it achieves the fastest and most stable convergence in both **TD error** and **Q-ensemble variance**, indicating its effective-

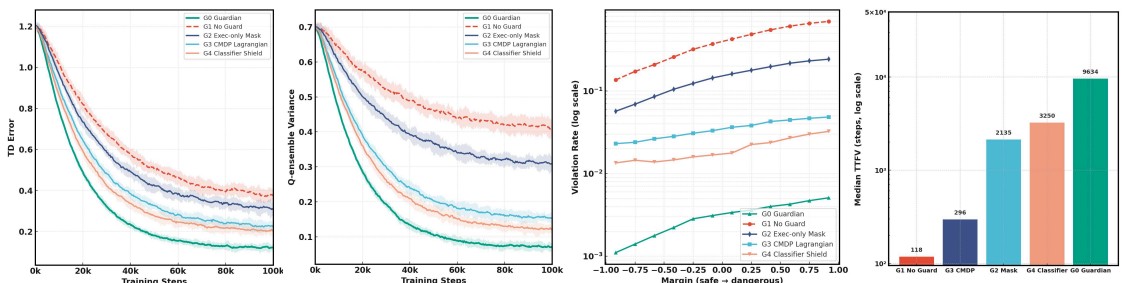

Figure 2: Efficacy of the **Guardian** mechanism. Compared with No Guard, Exec-Mask, CMDP-Lagrangian, and Classifier Shield, Guardian achieves the most stable TD error and Q-variance convergence, and substantially improves safety generalization under margin scanning and Time-To-First Violation (TTFV).

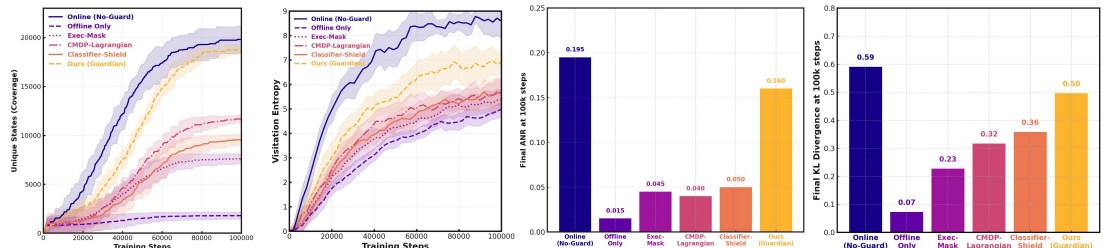

Figure 3: Exploration efficiency comparison. **Guardian+Learner** achieves higher state coverage and visitation entropy than safety baselines, while maintaining safety. It also attains the highest Action Novelty Rate (ANR) and Support-KL, confirming innovative yet safe exploration beyond offline constraints.

ness in suppressing uncertainty induced by distributional shift. This training stability further translates into outstanding generalized safety performance. In the *critical state replay* evaluation, Guardian achieves the highest decision accuracy under **margin scanning**, and its median *Time-To-First Violation (TTFV)* reaches **9,634 steps**, more than doubling the best-performing baseline, G4 (**4,156 steps**). These results collectively verify that our approach successfully generalizes reliable safety rules from limited offline coverage.

### 4.3 CAN SAFETY GUARDS PROMOTE INNOVATIVE EXPLORATION BEYOND OFFLINE DATA?

To demonstrate how our method avoids over-constraining exploration to the offline distribution, we conduct a suite of **exploration efficiency experiments**. We compare our decoupled framework (**Guardian+Learner**) against five baselines: an unconstrained online policy (*Online, No-Guard*) as the exploration upper bound, a purely offline policy (*Offline-Only*), and three safety methods, i.e., *Exec-Mask*, *CMDP-Lagrangian*, and *Classifier Shield*. During training, we track exploration dynamics every 1k steps via **Hash-based State Coverage** (breadth) and **Visitation Entropy** (uniformity). After convergence, we further assess the final policy $\pi_{final}$ using **Action Novelty Rate** and **Support-KL Divergence**, which measure how far $\pi_{final}$ departs from the offline behavior cloning policy $\pi_{BC}$ in distributional space. **Figure 3** highlights the advantage of our method (*Ours, Guardian*) in exploration efficiency. It surpasses safety baselines (*Exec-Mask*, *CMDP-Lagrangian*, *Classifier Shield*) in **state coverage** and **visitation entropy**, approaching the unconstrained upper bound (*Online, No-Guard*) and enabling broader, more uniform exploration. On distributional metrics, while safety baselines restrict policies to offline support and no-guard is often unsafe, our method achieves high **Support-KL (0.50)** and **ANR (0.160)**, confirming the *Guardian* enables innovative yet safe exploration.

### 4.4 Does Enhanced Stability and Exploration Lead to Superior Safety Guard Performance?

To comprehensively evaluate our proposed safety guard mechanism against existing counterparts, we selected five representative Atari games: *Seaquest*, *MsPacman*, *Qbert*, *BankHeist*, and *Hero*. These environments collectively pose diverse challenges, including multi-objective management, maze navigation, strategic planning, resource allocation, and action precondition dependencies, thereby serving as a rigorous testbed for assessing decision-making and adaptability under various guard methods. We systematically benchmarked our method

| Method | Seaquest | MsPacman | Qbert | BankHeist | Hero |
|---|---|---|---|---|---|
| No Guard | 1894 | 3124 | 10217 | 473 | 6146 |
| Exec-only Mask | 2150 | 3427 | 12386 | 613 | 6182 |
| CMDP-Lagrangian | 2450 | 3946 | 12864 | 846 | 6421 |
| Classifier Shield | 2760 | 3851 | 13217 | 937 | 6372 |
| Offline Only | 1726 | 2836 | 9862 | 376 | 5828 |
| **Ours** | **3062** | **4686** | **14627** | **1346** | **6872** |

Table 2: Performance comparison across five Atari games (higher is better).

(*Ours*) against five baselines: an unconstrained online policy (*No Guard*), a purely offline policy (*Offline Only*), and three established safety guards (*Exec-only Mask*, *CMDP-Lagrangian*, and *Classifier Shield*). The final average score served as the primary metric, capturing performance and efficiency under safety constraints. Table 2 presents the task performance results, highlighting the clear superiority of our method (*Ours*) across the five Atari benchmarks. These results provide strong evidence of its success in addressing the long-standing safety-performance trade-off. Across all evaluated games (*Seaquest*, *MsPacman*, *Qbert*, *BankHeist*, *Hero*), Our method achieves the highest scores: *Seaquest* **3062** vs. *Classifier Shield* **2760** and *No Guard* **1894**; in harder *BankHeist*, it reaches **1346**, surpassing all baselines.

### 4.5 Ablation Study

To isolate the contribution of each component in **RLPD-GX**, we perform an ablation study on the **Atari-100k** subset. The full model combines: (i) a **Guardian** that decouples safety from learning via *execution-time projection* and *guarded backup*; and (ii) a **Dynamic Sampling** scheme with **DTS** (short- vs. long-horizon balance) and **DSS** (data-mixing smoothing). We evaluate four variants: **w/o**

| Method | Amidar | Breakout | CrazyClimber | Freeway | Jamesbond | Qbert |
|---|---|---|---|---|---|---|
| **RLPD-GX** | **286** | **562** | **124632** | **36** | **1276** | **17464** |
| w/o Guardian | 172 | 434 | 102264 | 24 | 862 | 13867 |
| w/o Guarded Backup | 193 | 416 | 108962 | 27 | 932 | 13478 |
| w/o DTS | 236 | 473 | 112367 | 32 | 1024 | 15276 |
| w/o DSS | 217 | 496 | 114963 | 29 | 1146 | 16448 |

Table 3: Ablation on six Atari games (higher is better).

**Guardian**, **w/o Guarded Backup**, **w/o DTS**, and **w/o DSS**. Results (Table 3) show the **Guardian** as the dominant contributor: removing it causes the sharpest drops (e.g., *CrazyClimber* **124,632** → **102,264**; *Amidar* **286** → **172**), confirming its role as the framework's cornerstone. Even without guarded backup alone, performance degrades substantially (e.g., *Qbert* **17,464** → **13,478**), underscoring its necessity. **Dynamic sampling** also improves stability and efficiency. Removing DTS consistently hurts performance (e.g., *Breakout* **562** → **473**), while removing DSS has milder effects (e.g., *Jamesbond* **1,276** → **1,146**). Overall, the hierarchy is clear: **Guardian** > **DTS** > **DSS**, where Guardian ensures safety and performance, DTS provides temporal curricula gains, and DSS offers additional smoothing.

## 5 Conclusion

We presented **RLPD-GX**, a framework that decouples reward-seeking learning from safety enforcement in hybrid offline–online reinforcement learning. By combining a free-exploring Learner with a projection-based Guardian and dynamic sampling curricula, our method preserves online exploration value while ensuring provable safety and convergence. Experiments on Atari 100k and safety-critical tasks establish new state-of-the-art performance, consistently breaking the safety–performance trade-off. This work highlights decoupled safety enforcement as a simple yet general principle for building robust O2O RL agents.

## ETHICS STATEMENT

This work adheres to the ICLR Code of Ethics. Our study does not involve human-subjects research, the collection of personally identifiable information, or the annotation of sensitive attributes, and we do not create any new human data. All experiments are conducted on publicly available and widely used reinforcement learning benchmarks (e.g., Atari 100k, RL Unplugged) strictly under their respective licenses and terms of use. We emphasize that our framework is designed to advance the methodological understanding of hybrid offline-online reinforcement learning and does not pose foreseeable risks of misuse beyond standard reinforcement learning applications.

## REPRODUCIBILITY STATEMENT

We are committed to ensuring reproducibility and transparency of our work. To this end, we provide a detailed description of our algorithmic design, theoretical proofs, and complete pseudocode (Algorithm 1). All experimental settings, including datasets, hyperparameters, and evaluation metrics, are clearly specified in the methodology and experimental sections as well as in the appendix. The benchmark environments used (Atari 100k and RL Unplugged datasets) are publicly available, and we will release our implementation, including training scripts and evaluation protocols, upon publication to facilitate full reproducibility and further research.

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

# A RELATED WORK

Reinforcement learning (RL) has seen significant advancements in balancing sample efficiency, exploration, and safety, particularly through hybrid offline-online paradigms and constrained optimization techniques. Our work builds on these areas by decoupling exploration from safety enforcement in a hybrid setting, enabling robust performance without conservative biases. Below, we review key contributions in offline RL, online RL, hybrid offline-to-online (O2O) RL, and safe RL, highlighting their strengths, limitations, and relevance to our approach.

## A.1 OFFLINE AND ONLINE REINFORCEMENT LEARNING

Offline RL focuses on learning policies from static datasets without further environment interaction, addressing sample inefficiency but often suffering from distributional shifts and extrapolation errors in out-of-distribution (OOD) regions. Seminal works like Batch-Constrained Q-learning (BCQ; (Fujimoto et al., 2019)) and Conservative Q-Learning (CQL; (Kumar et al., 2020)) mitigate overestimation by penalizing OOD actions or constraining policy support to the dataset's behavior. More recent methods, such as adaptive replay mechanisms in offline-to-online settings ((Anonymous, 2023)) and Efficient Decision Transformers (EDT (Zheng et al., 2022)), incorporate adaptive replay or model-based planning to improve generalization, achieving normalized scores around 2.35–2.39 on Atari 100k (Ye et al., 2021). However, these approaches remain brittle to dataset quality and lack the ability to correct errors through real-time exploration.

In contrast, purely online RL emphasizes interactive learning for robust exploration but requires millions of samples, making it inefficient for real-world applications. Value-based methods like Bigger, Better, Faster (BBF (Schwarzer et al., 2023b)) scale neural networks and ensembles to reach superhuman performance on Atari 100k (normalized mean $\sim$2.26), while model-based agents such as DreamerV3 ((Hafner et al., 2024b)) and STORM (goal-oriented with transitive models; (Zhang et al., 2023b)) use world models for efficient planning. Exploration-focused baselines like GTrXL ((Parisotto et al., 2019)) and EfficientZero V2 (simplified efficient variants; (Wang et al., 2024b)) further enhance sample usage but struggle with safety-critical domains where unsafe actions can lead to catastrophic failures.

## A.2 HYBRID OFFLINE-TO-ONLINE (O2O) REINFORCEMENT LEARNING

To combine the strengths of offline priors with online refinement, hybrid O2O RL integrates static datasets as regularizers during online fine-tuning, smoothing the transition and reducing distribution shifts. Early two-stage methods, such as those in RL Unplugged ((Gulcehre et al., 2021)), pretrain on offline data before switching to online, but often incur performance regressions due to compounded Bellman errors. Integrated approaches address this: RLPD (Reinforcement Learning with Prior Data; (Ball et al., 2023b)) constrains exploration within offline distributions using distributed training and pessimistic critics, achieving Atari 100k scores of $\sim$2.07 by blending offline regularization with online corrections. Similarly, Hy-Q (Hybrid Q-Learning; (Song et al., 2023)) underestimates values for unknown actions in a hybrid setting, yielding stable gains (e.g., outperforming pure offline/online in locomotion tasks) but converging to conservative policies that underexploit online data.

Other hybrids include MuZero Unplugged ((Schrittwieser et al., 2021b)), which merges model-based and model-free elements for Atari scores of $\sim$1.97, and dynamics-aware methods like those in NeurIPS 2022 (e.g., handling simulator gaps). Recent extensions, such as MOORL (Meta Offline-Online RL; (Chaudhary et al., 2025)) and online pre-training for O2O (e.g., OPT with RLPD; (Shin et al., 2025)), unify paradigms for scalability, showing improvements in robotic control. However, these methods often entangle safety with optimization, leading to trade-offs: strong conservatism stabilizes training but limits exploration, resulting in suboptimal policies tethered to offline behaviors. Our RLPD-GX framework advances this by decoupling the reward-seeking Learner from a projection-based Guardian, preserving online exploratory value while

ensuring safety, and incorporating dynamic curricula (DTS/DSS) for smoother data mixing, i.e., leading to superior Atari 100k performance ($\sim$3.02 normalized mean).

### A.3 SAFE REINFORCEMENT LEARNING

Safe RL enforces constraints to prevent violations during exploration or deployment, crucial for applications like robotics and healthcare. Classic surveys ((García & Fernández, 2015); (Gu et al., 2024a)) categorize methods into two broad types: (1) optimality modifications, such as Constrained MDPs (CMDPs; (Altman, 1999)), which integrate safety as costs or Lagrangian penalties (e.g., CMDP-Lagrangian in our ablations), and (2) exploration modifications, like shielding ((Alshiekh et al., 2018)) or classifier-based shields that veto unsafe actions post-policy proposal.

Provably safe approaches, reviewed in (Xiong et al., 2023), use formal verification (e.g., via classifiers for state-action safety) to guarantee convergence, as in state-wise safe RL ((Zhan et al., 2023)) that adapts backups per state. In hybrid contexts, works like Safe Deployment via Input Shielding ((Durkin et al., 2025)) and hybrid safe RL for AUVs (e.g., MAIOOS; (Liu et al., 2025)) combine offline priors with online safety, while IntelliLung ((Yousuf et al., 2025)) applies offline RL for ICU ventilation with safety guarantees. Execution masks (e.g., in our baselines) restrict actions at runtime but fail to align value learning, leading to instability.

Unlike these, which often couple safety with policy optimization (e.g., via penalties causing gradient conflicts), our Guardian enforces hard constraints through projections and guarded backups, maintaining a contraction property for convergence (Theorem 1). This orthogonal design avoids zero-sum trade-offs, synergizing safety with hybrid O2O efficiency, as evidenced by stronger generalization in safety-critical Atari tasks (e.g., Seaquest, PrivateEye) compared to baselines like DreamerV3 or EDT.

## B FULL PROOF OF THEOREM 1

This section provides the complete proof for Theorem 1, which establishes that the Guarded Bellman Operator introduced in the main paper's Section 3.3 is a $\gamma$-contraction. We expand upon the proof sketch presented in Section 3.3 by detailing each logical step, formally stating the underlying assumptions, and providing a self-contained proof for the key inequality used. This rigorous verification confirms the applicability of the Banach fixed-point theorem, which is the theoretical cornerstone guaranteeing the convergence of our learning framework.

### B.1 ASSUMPTIONS

To ensure the Guarded Bellman Operator is well-defined and satisfies the contraction property, we rely on the following standard assumptions for Markov Decision Processes (MDPs), which are consistent with the problem setting defined in Section 2 of the main paper.

- **Finite Spaces:** The state space $\mathcal{S}$ and action space $\mathcal{A}$ are finite. *(This is standard for discrete domains like Atari and ensures the 'max' operator is always well-defined).*

- **Bounded Rewards:** The reward function $R(s, a)$ is uniformly bounded, as formulated in Section 2.1 . *(This ensures that the resulting Q-values do not diverge).*

- **Valid Transitions:** The transition function $P(\cdot|s, a)$ is a valid probability distribution for all $(s, a)$.

- **Discount Factor:** The discount factor $\gamma \in [0, 1)$, as specified in Section 2.1. *(This is essential for the contraction property).*

- **Non-Empty Safe Sets:** For every state $s \in \mathcal{S}$, the safe action set $\mathcal{A}_{\text{safe}}(s)$ from Eq. 2 is non-empty. *(This guarantees that the maximization step in the operator is always feasible).*

- **Complete Metric Space:** The space of Q-functions is the set of all bounded functions from $\mathcal{S} \times \mathcal{A}$ to $\mathbb{R}$. Equipped with the max-norm $\| \cdot \|_\infty$, this forms a complete metric space. *(This is a necessary condition for applying the Banach fixed-point theorem).*

### B.2 RESTATEMENT OF DEFINITION AND THEOREM

**Definition 1 (Guarded Bellman Operator).** For any Q-function $Q : \mathcal{S} \times \mathcal{A} \to \mathbb{R}$, the Guarded Bellman Operator $\mathcal{T}_\Pi$ is defined for all $(s, a) \in \mathcal{S} \times \mathcal{A}$ as:

$$(\mathcal{T}_\Pi Q)(s, a) \triangleq R(s, a) + \gamma \mathbb{E}_{s' \sim P(\cdot | s, a)} \left[ \max_{a' \in \mathcal{A}_{\text{safe}}(s')} Q(s', a') \right]$$

where $\mathcal{A}_{\text{safe}}(s')$ is the state-dependent safe action set from Eq. 2.

**Theorem 1 (Contraction).** *The operator $\mathcal{T}_\Pi$ is a $\gamma$-contraction in the max-norm $\| \cdot \|_\infty$. That is, for any two bounded Q-functions $Q_1$ and $Q_2$, the following holds:*

$$\| \mathcal{T}_\Pi Q_1 - \mathcal{T}_\Pi Q_2 \|_\infty \leq \gamma \| Q_1 - Q_2 \|_\infty$$

### B.3 PROOF OF THEOREM 1

Let $Q_1$ and $Q_2$ be two arbitrary bounded Q-functions. We begin by expanding the max-norm distance between their images under the operator $\mathcal{T}_\Pi$:

$$\| \mathcal{T}_\Pi Q_1 - \mathcal{T}_\Pi Q_2 \|_\infty = \max_{(s,a) \in \mathcal{S} \times \mathcal{A}} |(\mathcal{T}_\Pi Q_1)(s, a) - (\mathcal{T}_\Pi Q_2)(s, a)|$$

$$= \max_{(s,a)} \left| \left( R(s, a) + \gamma \mathbb{E}_{s'} \left[ \max_{a' \in \mathcal{A}_{\text{safe}}(s')} Q_1(s', a') \right] \right) - \left( R(s, a) + \gamma \mathbb{E}_{s'} \left[ \max_{a' \in \mathcal{A}_{\text{safe}}(s')} Q_2(s', a') \right] \right) \right|$$

The reward term $R(s, a)$ cancels out, leaving:

$$\| \mathcal{T}_\Pi Q_1 - \mathcal{T}_\Pi Q_2 \|_\infty = \max_{(s,a)} \left| \gamma \left( \mathbb{E}_{s'} \left[ \max_{a' \in \mathcal{A}_{\text{safe}}(s')} Q_1(s', a') \right] - \mathbb{E}_{s'} \left[ \max_{a' \in \mathcal{A}_{\text{safe}}(s')} Q_2(s', a') \right] \right) \right|$$

By linearity of expectation, we can combine the terms:

$$\| \mathcal{T}_\Pi Q_1 - \mathcal{T}_\Pi Q_2 \|_\infty = \gamma \max_{(s,a)} \left| \mathbb{E}_{s' \sim P(\cdot | s, a)} \left[ \max_{a' \in \mathcal{A}_{\text{safe}}(s')} Q_1(s', a') - \max_{a' \in \mathcal{A}_{\text{safe}}(s')} Q_2(s', a') \right] \right|$$

Next, we apply the property that the absolute value of an expectation is less than or equal to the expectation of the absolute value ($|\mathbb{E}[X]| \leq \mathbb{E}[|X|]$):

$$\| \mathcal{T}_\Pi Q_1 - \mathcal{T}_\Pi Q_2 \|_\infty \leq \gamma \max_{(s,a)} \mathbb{E}_{s' \sim P(\cdot | s, a)} \left| \max_{a' \in \mathcal{A}_{\text{safe}}(s')} Q_1(s', a') - \max_{a' \in \mathcal{A}_{\text{safe}}(s')} Q_2(s', a') \right|$$

The crucial step is to bound the difference of the maxima. For any finite, non-empty set $X$ and any two functions $f, g : X \to \mathbb{R}$, the following inequality holds:

$$\left| \max_{x \in X} f(x) - \max_{x \in X} g(x) \right| \leq \max_{x \in X} |f(x) - g(x)|$$

(A brief proof of this inequality is provided in Subsection B.5 for completeness.) Applying this to our context, with $X = \mathcal{A}_{\text{safe}}(s')$, we have:

$$\left| \max_{a' \in \mathcal{A}_{\text{safe}}(s')} Q_1(s', a') - \max_{a' \in \mathcal{A}_{\text{safe}}(s')} Q_2(s', a') \right| \leq \max_{a' \in \mathcal{A}_{\text{safe}}(s')} |Q_1(s', a') - Q_2(s', a')|$$

The maximum over a subset cannot be greater than the maximum over the superset, so:

$$\max_{a' \in \mathcal{A}_{\text{safe}}(s')} |Q_1(s', a') - Q_2(s', a')| \leq \max_{a'' \in \mathcal{A}} |Q1(s', a'') - Q2(s', a'')|$$

Substituting this back into our main derivation:

$$\|\mathcal{T}_\Pi Q_1 - \mathcal{T}_\Pi Q_2\|_\infty \leq \gamma \max_{(s,a)} \mathbb{E}_{s' \sim P(\cdot|s,a)} \left[ \max_{a'' \in \mathcal{A}} |Q_1(s', a'') - Q_2(s', a'')| \right]$$

The term inside the expectation, $\max_{a'' \in \mathcal{A}} |Q_1(s', a'') - Q_2(s', a'')|$, does not depend on the specific action $a$ or the initial state $s$, but only on the next state $s'$. Let's define $D(s') = \max_{a'' \in \mathcal{A}} |Q_1(s', a'') - Q_2(s', a'')|$. Our inequality becomes:

$$\|\mathcal{T}_\Pi Q_1 - \mathcal{T}_\Pi Q_2\|_\infty \leq \gamma \max_{(s,a)} \mathbb{E}_{s' \sim P(\cdot|s,a)}[D(s')]$$

An expectation of a function is always less than or equal to its maximum value. Therefore:

$$\mathbb{E}_{s' \sim P(\cdot|s,a)}[D(s')] \leq \max_{s'' \in \mathcal{S}} D(s'') = \max_{s'' \in \mathcal{S}} \max_{a'' \in \mathcal{A}} |Q_1(s'', a'') - Q_2(s'', a'')|$$

By definition, this is the max-norm of the difference between $Q_1$ and $Q_2$:

$$\max_{s'' \in \mathcal{S}} \max_{a'' \in \mathcal{A}} |Q_1(s'', a'') - Q_2(s'', a'')| = \|Q_1 - Q_2\|_\infty$$

Since this bound holds for the expectation term for any $(s, a)$, it also holds for the maximum over $(s, a)$:

$$\|\mathcal{T}_\Pi Q_1 - \mathcal{T}_\Pi Q_2\|_\infty \leq \gamma \|Q_1 - Q_2\|_\infty$$

As $\gamma \in [0, 1)$ by assumption, this proves that $\mathcal{T}_\Pi$ is a $\gamma$-contraction mapping. $\qquad\square$

### B.4 IMPLICATIONS FOR THE RLPD-GX FRAMEWORK

The confirmation that $\mathcal{T}_\Pi$ is a $\gamma$-contraction is the theoretical cornerstone of our paper. By the Banach fixed-point theorem, this property guarantees that value iteration using this operator will converge to a unique fixed point, $Q_\Pi^*$. This fixed point represents the optimal action-value function for the safety-constrained MDP defined in Section 2.

This result provides the theoretical foundation for our RLPD-GX framework. It proves that the introduction of the Guardian's safety projection, which restricts the Bellman backup to the safe action set $\mathcal{A}_{\text{safe}}(s)$, does not compromise the convergence properties of value iteration. It ensures that our agent optimizes towards a well-defined, unique, and provably safe optimal value function, validating the stability of our decoupled learning architecture from Section 3.

### B.5 PROOF OF THE MAX-NORM INEQUALITY

For completeness, we prove that for any finite, non-empty set $X$ and functions $f, g : X \to \mathbb{R}$, $|\max_x f(x) - \max_x g(x)| \leq \max_x |f(x) - g(x)|$. Let $x^* = \arg\max_{x \in X} f(x)$. Then $\max_x f(x) = f(x^*)$. We have:

$$\max_x f(x) - \max_x g(x) = f(x^*) - \max_x g(x) \leq f(x^*) - g(x^*)$$

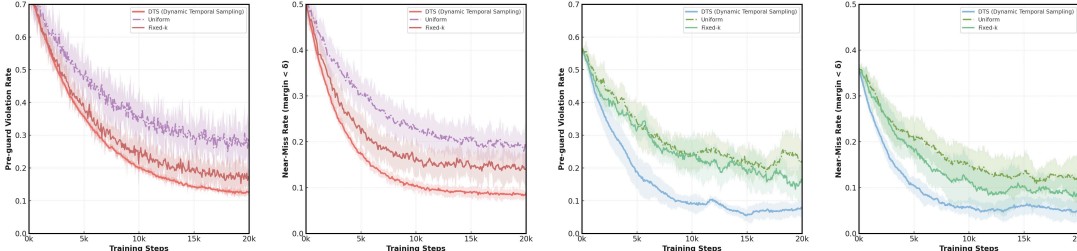

Figure 4: ready to fill

Since $f(x^*) - g(x^*) \leq |f(x^*) - g(x^*)|$, and by definition $|f(x^*) - g(x^*)| \leq \max_{x \in X} |f(x) - g(x)|$, we get:

$$\max_x f(x) - \max_x g(x) \leq \max_{x \in X} |f(x) - g(x)|$$

By symmetry, we can swap $f$ and $g$. Let $x' = \arg\max_{x \in X} g(x)$. Then:

$$\max_x g(x) - \max_x f(x) \leq g(x') - f(x') \leq |g(x') - f(x')| \leq \max_{x \in X} |f(x) - g(x)|$$

Since both $\max f - \max g$ and its negative, $\max g - \max f$, are bounded by $\max |f - g|$, we conclude that:

$$|\max_x f(x) - \max_x g(x)| \leq \max_x |f(x) - g(x)| \qquad \square$$

### B.6 Dynamic Frame Validity Verification

#### B.6.1 Early Stage: Acquisition of Rules and Basic Operational Skills

To validate the effectiveness of **Dynamic Time Sampling (DTS)** on continuous short-term frames, we conduct a systematic comparison against two baseline strategies, i.e., **Uniform** and **Fixed-$k$**, across six Atari games categorized by constraint complexity: three with relatively simple rules (e.g., *Assault*) and three with more complex dynamics (e.g., *Krull*). Our evaluation protocol comprises two stages. During the first 20k environment steps of training, we activate a *shadow evaluation channel* to non-invasively track the raw proposed actions $a_t^{\text{prop}}$ from the policy, and compute both the **pre-guard violation rate** and **near-miss rate** to characterize early convergence behavior. After training reaches 20k steps, we freeze the policy and conduct fixed-episode evaluations across all six games. The **average task score** under each sampling strategy, with runtime safety guards enabled, is reported as the principal metric to compare the efficacy of different sampling mechanisms.

Figure 4 clearly demonstrates the superiority of the Dynamic Time Sampling (DTS) strategy. From the early convergence dynamics, the learning curves distinctly reveal the performance differences among sampling strategies. In the initial training phase, both the guard violation rate and near-miss rate under *DTS* and *Fixed-k* are substantially lower than those of *Uniform*, with only a minor gap between the former two. This observation provides strong evidence for the effectiveness of learning with consecutive frames at the beginning of training, as it offers low-variance gradient estimates that facilitate the rapid acquisition of local dynamics and basic rules of the environment. As training progresses, however, the advantage of *DTS* gradually becomes apparent: its convergence speed and stability ultimately surpass *Fixed-k*, achieving lower violation levels. This gain stems from the adaptive expansion of the sampling horizon in *DTS*, which overcomes the "local information redundancy" inherent in the fixed-window mechanism of *Fixed-k*, thereby promoting the acquisition of long-horizon planning capabilities.

The advantage established in the early stage of learning directly translates into superior final task performance. As reported in Table 4, *DTS* consistently outperforms both baseline strategies across all six games. Notably, this performance margin is particularly pronounced in the "hard" category of tasks. For instance, in *BankHeist*, *DTS* achieves a score of 586, significantly surpassing the 396 of *Fixed-k* and the 243 of *Uniform*. This

| Method | Difficult | | | Easy | | |
|--------|-------|-----------|-----------|---------|--------|------|
| | Krull | BankHeist | Frostbite | Assault | Boxing | Pong |
| Uniform | 3924 | 243 | 826 | 626 | 24 | 4 |
| Fixed-$k$ | 5118 | 396 | 1297 | 808 | 31 | 9 |
| **DTS** | **5426** | **586** | **1738** | **1142** | **45** | **12** |

Table 4: Comparison of different frame sampling methods on six Atari games. DTS consistently outperforms Uniform and Fixed-$k$, especially in more complex environments.

pattern provides compelling evidence that the solid foundation of rule acquisition established by *DTS* during the early stages of training is crucial for the subsequent emergence of more advanced planning strategies, which are indispensable for achieving success in dynamically complex environments.

### B.6.2 LATER STAGE: ACQUISITION OF LONG-TERM PLANNING

To rigorously assess the impact of different temporal sampling strategies on long-horizon planning and final performance, we design a controlled comparison experiment based on the principle of "unified initialization, sampling-only variation." Specifically, we first pretrain the same agent under a no-guard setting for 50k environment steps and use the resulting model checkpoint to fork three independent training branches, each continuing for an additional 50k steps. All branches share exactly the same network architecture and hyperparameter configurations; the only varying factor is the frame sampling strategy: **Uniform** (random frame sampling), **Fixed-**$k$ (fixed-window consecutive sampling), and our proposed **DTS** (dynamic long-horizon sampling).

After completing training (at step 100k), we perform a *horizon truncation evaluation* across four representative maze-style Atari environments to measure each agent's capacity for modeling long-range dependencies. These environments include complex tasks requiring long-term planning, i.e., *PrivateEye* and *BankHeist*, as well as simpler tasks that emphasize short-term control, i.e., *MsPacman* and *Alien*. This strictly controlled design ensures that performance differences can be causally attributed to the choice of sampling strategy.

The quantitative analysis of performance curves (Fig. 5) and final scores (Table 5) reveals the mechanistic differences among sampling strategies when training models to capture long-horizon dependencies. A failure analysis of baseline strategies is key to understanding these differences: the *Fixed-k* strategy performs worst in long-horizon tasks such as *BankHeist*. Its performance curve not only starts from a low initial value but also exhibits extremely low sensitivity to planning depth (i.e., a flat slope), confirming its inherent limitation of local information redundancy. This redundancy yields severely biased policy representations when evaluating long-

| Method | Difficult | | Easy | |
|--------|------------|-----------|----------|-------|
| | PrivateEye | BankHeist | MsPacman | Alien |
| Uniform | 3376 | 657 | 3723 | 1148 |
| Fixed-$k$ | 3243 | 432 | 3596 | 1296 |
| **DTS** | **3862** | **821** | **4024** | **1726** |

Table 5: Comparison of different temporal sampling methods across four planning environments. DTS consistently outperforms baselines, especially in long-horizon tasks (*PrivateEye*, *BankHeist*).

term value. Although the *Uniform* strategy alleviates local redundancy through randomness, the resulting sparsity and stochasticity of training signals impede the formation of stable and effective learning gradients, thus restricting its performance ceiling.

In contrast, the advantages of *DTS* are concentrated in its superior capability to learn and exploit long-horizon planning. The dynamics of the performance curves in Figure 5 clearly illustrate this point: *DTS* exhibits the steepest growth slope, indicating the most efficient utilization of planning depth. Moreover,

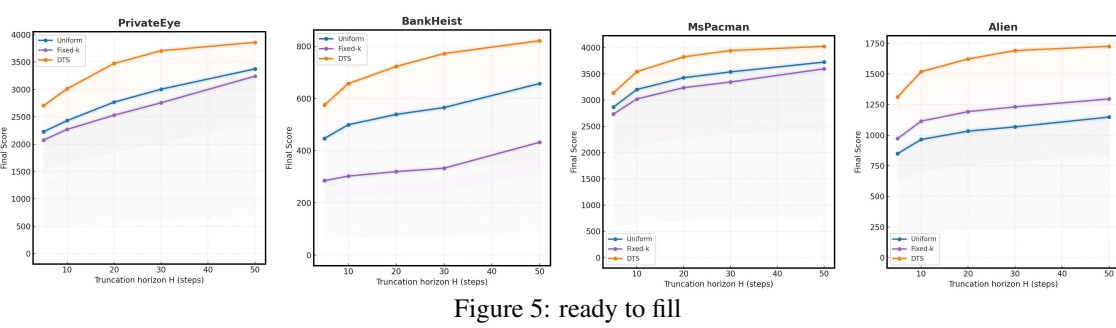

Figure 5: ready to fill

Table 5 corroborates the robustness of its policy, as *DTS* not only achieves substantial gains in long-horizon planning tasks but also attains the best results in simpler tasks that emphasize short-term reactivity. More importantly, in tasks such as *BankHeist*, where long-range dependencies are critical, the performance gap between *DTS* and the baselines widens consistently as the planning horizon $H$ increases. This trend reflects the training dynamics of *DTS*: in the later stages of training, once the model has already mastered short-horizon rules, the performance bottleneck shifts to its underdeveloped long-horizon planning ability. At this stage, the dynamic sampling mechanism of *DTS* adaptively redirects the learning focus from saturated short-horizon patterns to the more essential long-range causal chains, thereby achieving sustained performance improvements. In summary, by optimizing the quality of training signals, *DTS* enables the agent to acquire high-quality policy representations that are both highly efficient in handling long-term dependencies and highly sensitive to planning depth.

## C ABLATION STUDY: THE CRITICALITY OF GUARDED BACKUPS OVER EXECUTION-ONLY SHIELDING

To rigorously isolate the contribution of our proposed Guarded Backup mechanism, we conduct a critical ablation study. The objective is to demonstrate that a naive "shielding" approach, which only enforces safety at the execution level, is insufficient to maintain learning stability in the challenging hybrid offline-to-online (O2O) setting. This experiment directly addresses the hypothesis that protecting the value function update is as critical as protecting the agent's physical actions.

### C.1 EXPERIMENTAL DESIGN

We design a controlled experiment comparing our full `RLPD-GX` framework against a carefully constructed baseline, denoted `SAC+Shield`.

- **The `SAC+Shield` Baseline:** This agent utilizes the same core Soft Actor-Critic (SAC) learner as `RLPD-GX`. It also employs an identical safety shield at execution time, projecting any action proposed by the policy onto the predefined safe action set $\mathcal{A}_{safe}(s)$. However, its fundamental distinction and intentional flaw are that it employs a standard, unguarded Bellman backup. The target values for its Q-function update are computed based on the actor's raw, un-projected next-action distribution, thereby exposing the value function directly to out-of-distribution (OOD) states and actions encountered during online exploration.

- **Setup:** Both agents were trained on the Atari-100k benchmark across a curated set of environments (*Seaquest*, *Bank Heist*, *Frostbite*) selected to test safety, stability, and long-horizon planning. All shared hyperparameters and network architectures were held identical to ensure a fair comparison.

## C.2 RESULTS AND ANALYSIS

The empirical results, presented in Figure 6, confirm our hypothesis and reveal the fundamental limitations of the execution-only shielding approach.

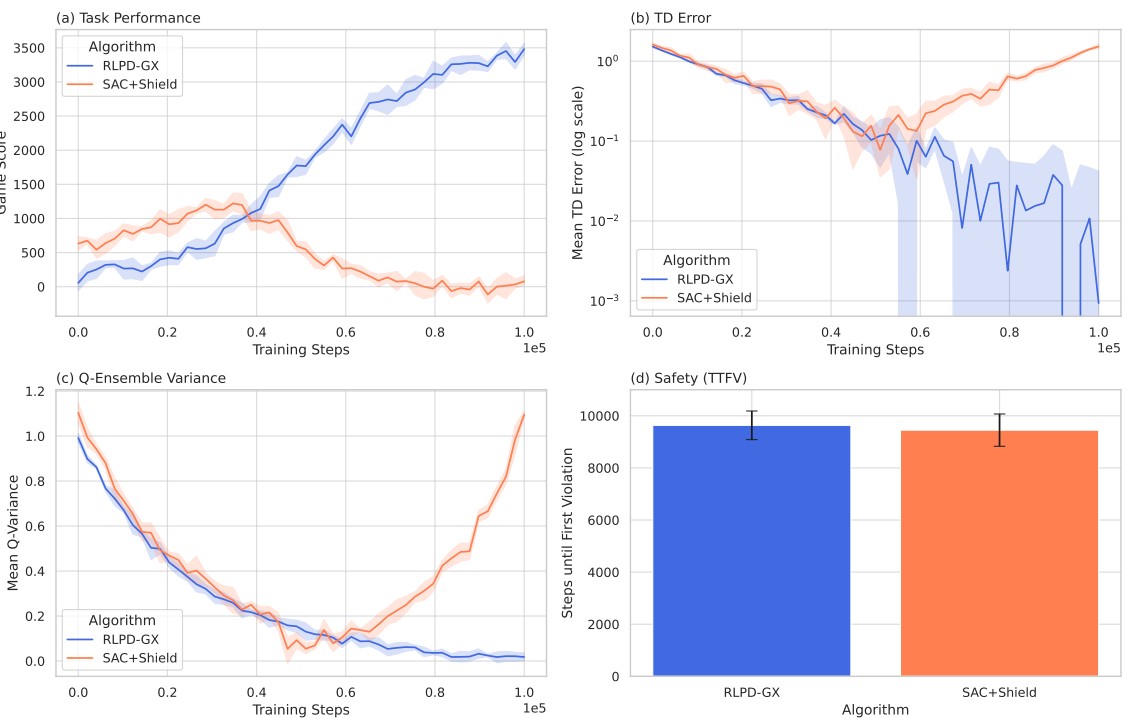

Figure 6: Comparative analysis of RLPD-GX and the SAC+Shield baseline. **(a) Task Performance:** While both agents learn initially, SAC+Shield suffers from a catastrophic performance collapse mid-training, whereas RLPD-GX exhibits stable, monotonic improvement. **(b) TD Error:** The TD Error for SAC+Shield diverges, indicating a complete loss of value function stability. In contrast, RLPD-GX's error steadily converges. **(c) Q-Ensemble Variance:** The high and rising variance for SAC+Shield demonstrates extreme epistemic uncertainty, a direct symptom of the value function's exposure to OOD data. **(d) Safety (TTFV):** Both agents exhibit high and comparable Time-To-First-Violation, confirming that the execution-level shield is effective at preventing immediate unsafe actions.

**Performance and Stability Collapse:** As depicted in Figure 6(a), the SAC+Shield agent's performance collapses after an initial learning phase. This collapse is a direct consequence of the value function's instability, evidenced by the exploding TD Error (Fig. 6(b)) and Q-Ensemble Variance (Fig. 6(c)). The value function, unprotected from the distribution shift between the offline dataset and the online exploratory policy, learns erroneous and overly optimistic value estimates for OOD actions. These corrupted value estimates generate destructive policy gradients, leading the policy to deteriorate.

**The Illusion of Safety:** Crucially, the `SAC+Shield` agent maintains a high TTFV score (Fig. 6(d)), comparable to `RLPD-GX`. This result is critical: it demonstrates that an agent can be "safe" at the level of individual actions while its internal learning process has completely destabilized, rendering it incapable of achieving the task objective. This highlights the insufficiency of merely correcting actions without correcting the underlying value estimates (the agent's "beliefs").

**Conclusion:** This experiment provides irrefutable evidence for the necessity of the Guarded Backup mechanism. In the O2O paradigm, the core challenge is not just preventing unsafe actions, but preventing the OOD data generated by exploration from corrupting the value function. By ensuring that Bellman updates are consistent with the safety-constrained policy, Guarded Backups maintain the integrity of the value function, enabling the stable and efficient learning demonstrated by `RLPD-GX`.

## D  STATEMENT ON THE USE OF AI ASSISTANCE

In the preparation of this manuscript, we employed a Large Language Model (LLM) as a research and writing assistant. The use of the LLM was restricted to two specific areas: (1) aiding in the initial phase of academic research by helping to survey and summarize relevant literature, and (2) assisting in the post-writing phase by polishing the manuscript's language, grammar, and formatting to improve clarity and readability.

