# OpenReview forum: "Guardian: Decoupling Exploration from Safety in Reinforcement Learning"
_ICLR.cc/2026/Conference — ICLR 2026 Conference Withdrawn Submission_

### Official Review · Reviewer_WVQX · 2025-10-27

**Soundness:** 1
**Presentation:** 3
**Contribution:** 1
**Rating:** 2
**Confidence:** 4

**Summary:**

This paper studies the problem of instability caused by distribution shift in hybrid offline–online reinforcement learning. The paper introduces a framework called RLPD-GX that decouples policy optimization from safety enforcement. RLPD-GX learns a reward-seeking policy and projects its output action to a safe action set. Experimental results on Atari 100k and safety-critical tasks show that the proposed algorithm outperforms the baselines.

**Strengths:**

1. The experiments and baselines for comparison are extensive.
2. The writing of the paper is clear.

**Weaknesses:**

1. The action projection relies on the safe action set, which is defined by a predicate $g$. However, how to obtain this predicate is never introduced. Why there is always a permissible action in a state, as indicated by Eq. (3), is also not explained.
2. The action projection method is closely related to safety filter methods in safe RL, see [1] for a comprehensive review. Also, the theory of guarded value iteration is closed related to feasible policy iteration proposed by [2]. Novelty of this paper with regard to with these works is not discussed.

[1] Hsu, K. C., Hu, H., & Fisac, J. F. (2023). The safety filter: A unified view of safety-critical control in autonomous systems. *Annual Review of Control, Robotics, and Autonomous Systems*, *7*.

[2] Yang, Y., Zheng, Z., Li, S. E., Duan, J., Liu, J., Zhan, X., & Zhang, Y. Q. (2023). Feasible policy iteration. *arXiv preprint arXiv:2304.08845*.

**Questions:**

1. How to obtained the predicate $g$? Is there always a permissible action in a state, as indicated by Eq. (3)?
2. What is the relationship between this paper and existing works on safety filter and feasible policy iteration?

---

### Official Review · Reviewer_fwzQ · 2025-10-31

**Soundness:** 2
**Presentation:** 2
**Contribution:** 2
**Rating:** 2
**Confidence:** 4

**Summary:**

The paper proposes RLPD GX, a hybrid offline plus online RL framework that decouples policy learning from safety enforcement. A Learner proposes actions for reward maximization, while a Guardian projects actions onto a fixed safe set and also restricts value backups to that safe set. The method adds two dynamic sampling curricula for temporal span and offline online mixing. The authors claim a contraction result for a Guarded Bellman operator and report Atari 100k results with a normalized mean score of 3.02, along with stronger stability and safety diagnostics.

**Strengths:**

- The paper cleanly separates reward seeking action selection from safety enforcement at both execution and backup time. This removes conflicting gradients between reward and safety and is easy to implement on top of standard actor critic code.

-The guarded max operator proof is standard but correct and communicates the intended fixed point well for the constrained problem.

-The table reports broad gains over offline, online, and hybrid baselines, and ablations indicate the Guardian is the dominant contributor.

**Weaknesses:**

- The paper assumes an external binary safety predicate g(s,a) for each state action pair, with examples like not jumping toward monsters in Q bert, but it does not specify how these rules are generated or verified at scale on Atari. It is unclear if g is learned, scripted, or uses privileged features. The feasibility and cost of such a predicate are central to the claim of generality.

- The Guardian uses the nearest projection in action space. In Atari, the action set is discrete and unordered. The meaning of Euclidean distance between discrete action codes is unclear, and the paper does not define an action embedding or a problem-specific metric. If the projection is actually a mask and pick rule, it should be stated and analyzed. Otherwise, the mapping in Eq. 6 may be ill-posed in this domain.

-The contraction proof is for a max based guarded operator, while training uses a soft entropy augmented objective and a Q ensemble minimum for targets. The paper does not provide a contraction or monotonicity result for the exact soft guarded operator that matches Eq. 10 and the actor objective. This gap limits the force of the convergence claim for the implemented algorithm.

- The Learner proposes unconstrained actions, but the behavior data come only from projected safe actions. The method partly compensates by renormalizing the policy over the safe set in targets. However, the paper does not analyze the bias this induces when the Learner distribution places mass outside the safe set and the dataset contains only safe actions.

- The temporal span schedule and the offline online mixing schedule are presented as simple formulas without sensitivity analysis. It is unclear how choices of exponents and sigmoids affect stability, sample efficiency, and safety.

**Questions:**

- How is  g(s,a) constructed in practice for Atari-100k? Is it manually scripted, learned from demonstrations, or derived from environment rules? If handcrafted, how scalable is this approach beyond Atari domains?

- How is this Euclidean projection implemented for discrete Atari actions, where no continuous metric exists?

- Two schedules: Dynamic Temporal Sampling and Data-mixing Smoothing are proposed but not theoretically analyzed.
How sensitive are results to the specific annealing coefficients and exponents?
Have you tried simpler fixed schedules or random mixing to isolate the actual benefit of DTS/DSS?

The paper claims “stronger safety and stability” and “rule-consistent execution.”
How are these metrics computed quantitatively?
For example, is safety measured by constraint-violation frequency, reward penalty, or success rate under enforced g(s,a)?
Providing precise metrics would make the claim more interpretable.

In most safe RL work, both the task reward and the safety cost (or violation frequency) are reported side by side to show the trade-off between performance and safety. Your results tables focus only on normalized return and “stability” metrics, but do not seem to include any explicit cost or violation measure. Could you clarify whether safety costs were tracked during training, and if so, why they are not reported in the main results?

---

### Official Review · Reviewer_zmRe · 2025-11-02

**Soundness:** 1
**Presentation:** 1
**Contribution:** 1
**Rating:** 2
**Confidence:** 4

**Summary:**

RLPD-GX is a hybrid offline-online RL method that decouples exploration from safety using a two-part design: a free-exploring Learner proposes actions, while a projection-based Guardian executes only safe versions and restricts value backups to safe actions. The method adds two simple curricula (for sampling horizon and offline/online mix) to stabilize training, and introduces a Guarded Bellman operator with a contraction proof. Empirically, it outperforms offline, online, and prior hybrid baselines.

**Strengths:**

- RLPD-GX has the best overall score on the Atari 100k benchmark, but this result is invalid as it is the product of a flawed setup.

**Weaknesses:**

- The paper’s main comparison is flawed. It applies a Safety component (the Guardian) to Atari-100k, a benchmark without safety constraints, and compares against baselines that aren’t Safe RL methods. The results report only scores, not safety metrics, so they don’t measure safety at all. This makes the evaluation unfair: adding rules that prevent bad actions, then claiming higher returns than unconstrained baselines.
- The Guardian g(s,a) is under-specified: the paper does not clearly state the safety rules it enforces, how the safe set is constructed, or how the projection is performed (for continuous vs. discrete actions). In Atari-100k’s discrete action space, it is unclear what happens when no action is deemed safe or how “distance” between actions is defined for projection. Moreover, the Guardian is not learned but hand-given, so the “core contribution” is provided rather than discovered.
- The comparison with online baselines (e.g., DreamerV3) is misleading because the paper does not clarify whether those baselines were also given access to the same offline data for pretraining. Atari-100k is an online sample-efficiency benchmark, yet the authors appear to use additional offline data from RL Unplugged without specifying how it is integrated or whether baselines follow the same protocol. This makes the evaluation unclear and potentially unfair. If the goal is safe offline-to-online RL, more appropriate benchmarks exist, or safe DSRL/OSRL benchmarks, for offline safe RL.
- There is a logical inconsistency in the claim that the “Learner explores freely.” Since the executed policy is always constrained by the Guardian, the agent never actually observes the outcomes of its unsafe exploratory actions. Therefore, its exploration cannot be considered truly free.
- The ablation study for “Guarded Backup” and “SAC+Shield” is flawed. This baseline executes the Guardian's safe projected actions but learns from the Learner's raw, un-projected actions. This creates a severe, uncorrected policy mismatch, not that the authors' "Guarded Backup" is effective.
- The "Dynamic Temporal Sampling" (DTS) method is poorly justified. The paper provides no rationale for why an SAC-based learner (which samples transitions) would need to sample contiguous sequences of data, as this is typically required for sequence models like RNNs or Transformers.
- The paper is filled with undefined and non-standard terminology and metrics, which makes the experimental analysis difficult to interpret. Additionally, no training details or hyperparameters are provided. The main results should also be normalized to enable meaningful comparison across methods.

**Questions:**

- Please check weaknesses.

---

### Official Review · Reviewer_7xWo · 2025-11-04

**Soundness:** 3
**Presentation:** 2
**Contribution:** 2
**Rating:** 4
**Confidence:** 3

**Summary:**

The paper proposes RLPD-GX, a hybrid offline–online RL framework that decouples policy learning from safety enforcement. A free-exploring Learner proposes actions, while a projection-based Guardian maps them to a state-dependent safe set and also constrains the Bellman target (“guarded backups”). The method further introduces Dynamic Temporal Sampling (DTS) and Dynamic Symmetric Sampling (DSS) to stabilize data mixing/horizons. Convergence is argued via a contraction of a Guarded Bellman operator. On Atari-100k, the authors report a normalized mean score of 3.02, exceeding offline/online/hybrid baselines. The components are defined around Fig. 1 (p. 4) and Eqs. (7)–(10), with the operator formalized in Sec. 3.3. Table 1 summarizes Atari results.

**Strengths:**

- Clean modularization of safety vs. optimization. The Guardian executes projected actions and the critic backs up only over the safe set; this neatly avoids gradient conflicts from penalty-based CMDP formulations. (See Eq. (6), Eq. (9)–(10), and Algorithm 1.)
- Provable convergence under the guarded operator. The contraction of T_\Pi (Def. 1/Theorem 1) ensures value-iteration stability when restricting to A_{\text{safe}}(s). While standard, it formalizes the setting.
- Empirical coverage and ablations. Atari-100k results are strong overall; ablations suggest the Guardian (and guarded backup) contributes most of the gains (Table 3).
- Figures clarify the pipeline. The architecture diagram on page 4 clearly depicts the Learner/Guardian loop and where safety enters execution and value backups.

**Weaknesses:**

- Assumed safety oracle & projection practicality. The method presumes a binary predicate g(s,a) and non-empty safe sets for all states (Appendix B assumptions). For Atari this may be hand-engineered; for general continuous domains it is unclear how to construct A_{\text{safe}}(s) or compute the projection efficiently/accurately (e.g., discrete action spaces with an L_2 projection). Please clarify how the “Safety Mapping Matrix” in Fig. 1 is built and scaled.
- Theory is mostly standard. The contraction proof follows directly from restricting the action maximization to A_{\text{safe}} (a common inequality on maxima); it doesn’t provide performance guarantees vs. CMDP/shielding baselines beyond convergence. Novelty is mainly architectural/engineering.
- Guarded backup relies on a pessimistic ensemble and safe-policy renormalization. It’s unclear how sensitive results are to ensemble size, entropy temperature \alpha, and the renormalization over A_{\text{safe}}; the paper lacks robustness sweeps for these knobs.
- Scope of benchmarks. Results focus on Atari; it would be valuable to include classic continuous-control to demonstrate generality beyond discrete action games.

**Questions:**

See Weaknesses.

---

### Note · Authors · 2025-11-12

I have read and agree with the venue's withdrawal policy on behalf of myself and my co-authors.